# Microtubules orchestrate local translation to enable cardiac growth

Emily A. Scarborough [1,7], Keita Uchida [1,7], Maria Vogel[1], Noa Erlitzki [1,2], Meghana Iyer[1,3], Sai Aung Phyo[1,4], Alexey Bogush[1], Izhak Kehat [5,6] & Benjamin L. Prosser [1✉]

Hypertension, exercise, and pregnancy are common triggers of cardiac remodeling, which occurs primarily through the hypertrophy of individual cardiomyocytes. During hypertrophy, stress-induced signal transduction increases cardiomyocyte transcription and translation, which promotes the addition of new contractile units through poorly understood mechanisms. The cardiomyocyte microtubule network is also implicated in hypertrophy, but via an unknown role. Here, we show that microtubules are indispensable for cardiac growth via spatiotemporal control of the translational machinery. We find that the microtubule motor Kinesin-1 distributes mRNAs and ribosomes along microtubule tracks to discrete domains within the cardiomyocyte. Upon hypertrophic stimulation, microtubules redistribute mRNAs and new protein synthesis to sites of growth at the cell periphery. If the microtubule network is disrupted, mRNAs and ribosomes collapse around the nucleus, which results in mis-localized protein synthesis, the rapid degradation of new proteins, and a failure of growth, despite normally increased translation rates. Together, these data indicate that mRNAs and ribosomes are actively transported to specific sites to facilitate local translation and assembly of contractile units, and suggest that properly localized translation – and not simply translation rate – is a critical determinant of cardiac hypertrophy. In this work, we find that microtubule based-transport is essential to couple augmented transcription and translation to productive cardiomyocyte growth during cardiac stress.

[1] Department of Physiology, Pennsylvania Muscle Institute, University of Pennsylvania Perelman School of Medicine, Philadelphia, PA, USA. [2] Graduate Program in Biochemistry and Molecular Biophysics, University of Pennsylvania Perelman School of Medicine, Philadelphia, PA, USA. [3] Department of Bioengineering, University of Pennsylvania, Philadelphia, PA, USA. [4] Graduate Program in Genetics and Epigenetics, University of Pennsylvania Perelman School of Medicine, Philadelphia, PA, USA. [5] The Rappaport Institute and the Bruce Rappaport Faculty of Medicine, Technion-Israel Institute of Technology, Haifa, Israel. [6] Department of Cardiology and the Clinical Research Institute at Rambam, Rambam Medical Center, Haifa, Israel. [7] These authors contributed equally: Emily A. Scarborough, Keita Uchida. ✉email: bpros@pennmedicine.upenn.edu

The heart remodels its size and shape in response to chronic increases in cardiac demand. In adults, this occurs almost exclusively through hypertrophy of existing cardiomyocytes. Cardiomyocyte growth, and the subsequent increase in organ mass, is driven largely by the addition of new sarcomeres, the basic contractile units of muscle. During pregnancy or exercise, physiological hypertrophy increases cardiomyocyte force production to match elevated demand, while during pathological remodeling, aberrant changes in cardiomyocyte structure can lead to contractile dysfunction and progression to heart failure.

The signal transduction pathways that initiate hypertrophy are well-defined. Broadly, mechanical or neurohumoral stimuli trigger receptor-dependent signaling cascades that activate transcription and translation to support new sarcomere formation[1–3]. Conversely, any post-transcriptional mechanisms that execute a growth program and govern new sarcomere addition remain opaque. Recent work indicates that mRNAs and ribosomes localize to the sarcomere, supporting a model of local sarcomere synthesis and assembly[4,5]. Yet the mechanisms that position the translational machinery, and whether such a mechanism may contribute to hypertrophy, is unknown.

The cardiomyocyte microtubule network is also implicated in cardiac hypertrophy. There is a proliferation, stabilization, and post-translational modification of microtubules concomitant with hypertrophy[6,7], and microtubule depolymerization can prevent pressure overload-induced hypertrophy in animal models[8–10]. As general regulators of intracellular trafficking, microtubules could impinge on many aspects of a hypertrophic pathway.

From the above, we hypothesized that active, microtubule-based transport may distribute mRNA and ribosomes throughout the cardiomyocyte, and that such spatiotemporal control of translation may be required for cardiac hypertrophy. We find that kinesin-1 dependent transport of mRNA and ribosomes along microtubule tracks is essential for localizing translation in the cardiomyocyte, and that this serves as a necessary bridge to couple increased protein synthesis to productive cardiac growth.

## Results

**Microtubules couple increased translation to cardiac growth.** Previous studies established that chronic colchicine treatment is sufficient to block pressure overload-induced cardiac hypertrophy[8–10]. We aimed to test if this effect is specific to hemodynamic overload, or if microtubule destabilization inhibits cardiac growth regardless of the upstream stimulus. To this end, we validated a mouse model of adrenergic stress-induced hypertrophy using phenylephrine (PE) (Fig. 1a), which activates α1-adrenergic receptors to stimulate cardiomyocyte protein synthesis[11]. To assess if PE drives microtubule proliferation, we performed a tubulin fractionation assay on the myocardial tissue of treated animals. In PE-treated mice, levels of polymerized tubulin increased robustly (~four-fold), while free (soluble) tubulin increased in most animals, but more variably (Fig. 1b, for all uncropped annotated blots, see supplementary material). Colchicine treatment alone depolymerized microtubules, but did not increase levels of free tubulin, consistent with tubulin mRNA autoregulation[8,12]. When co-administered, colchicine completely blocked the PE-induced increase in polymerized tubulin (Fig. 1b), while also decreasing total tubulin levels relative to both PBS and PE-injected mice (Supplementary Fig. 1).

PE caused significant hypertrophy, as measured by a consistent ~30% increase in the ratio of heart weight to tibia length (Fig. 1c). While colchicine alone had no effect on heart size, concurrent administration was sufficient to prevent PE-induced hypertrophy (Fig. 1c). Together with previous reports[8–10], the anti-hypertrophic effect of microtubule destabilization appears generalizable to common upstream stimuli (e.g., pressure overload, adrenergic stress), suggesting the disruption of a shared downstream mechanism of cardiac growth.

We next asked whether microtubule disruption may block hypertrophy by preventing stress-induced transcriptional changes and increased protein translation. We first quantified canonical transcriptional markers of hypertrophic stress (*Nppa*) and fetal reprogramming (decreased *Myh6*:*Myh7*) by qPCR. As expected, PE increased *Nppa* and *Myh7* expression, while reducing *Myh6* (Fig. 1d, Supplementary Fig. 1b). Surprisingly, PE elicited a similar, robust hypertrophic transcriptional signature in the presence of colchicine. We next determined rates of protein translation by injecting mice with puromycin (Fig. 1a), a tRNA analog that incorporates into nascent peptides and is detectable via western blot. PE similarly and significantly increased rates of myocardial protein synthesis in the presence or absence of colchicine (Fig. 1e), even though in the former, hearts did not hypertrophy (Fig. 1c). Together, these data indicate that microtubules couple an agonist-induced increase in protein translation to productive cardiac growth.

To interrogate potential mechanisms, we established an in vitro model of cardiac hypertrophy using nano-patterned neonatal rat ventricular myocytes (NRVMs, Fig. 2a). Plating NRVMs on nano-patterned substrates promotes a more adult-like morphology, increasing NRVM aspect ratio, sarcomere alignment (Supplementary Fig. 2a–c), and organization of the non-sarcomeric cytoskeleton[13]. First, we treated patterned NRVMs with either DMSO, 10 μM colchicine, 100 μM PE + DMSO, or 100 μM PE + 10 μM colchicine for 24 h and assessed changes in cell size through live-cell imaging of a lipophilic dye (Fig. 2b). We confirmed that the microtubule network was abolished in colchicine-treated conditions (Fig. 2b). PE alone significantly increased NRVM size (Fig. 2c), which was driven by lengthening along the long axis of the myocyte (i.e., eccentric hypertrophy, Supplementary Fig. 2d) in this patterned, confluent system. PE-induced hypertrophy was fully blocked by simultaneous colchicine treatment (Fig. 2c), complementing previous findings[10,14]. We repeated this experiment using a different hypertrophic agonist, isoproterenol, which activates β1 and β2 adrenergic receptors to increase protein synthesis. Isoproterenol (1 μM) induced a similar increase in NRVM size as PE treatment, and colchicine similarly prevented isoproterenol-induced growth (Supplementary Fig. 3). Finally, we performed the experiment with nocodazole, a microtubule depolymerizing agent that acts through a separate mechanism from colchicine, to control for off-target effects. Nocodazole (10 μM) also fully blocked PE-induced increases in NRVM size (Fig. 2c). Together these data indicate that microtubules are required for adrenergic stress-induced hypertrophy in NRVMs.

We next assessed protein translation rates and transcriptional signatures. After 24 h of DMSO, colchicine, PE + DMSO, or PE + colchicine treatment, NRVMs were pulsed with puromycin. As a negative control, cells were pre-treated with cycloheximide to block protein translation. PE-treated NRVMs showed an increase in puromycinylation regardless of the presence or absence of microtubules (Fig. 2d). Cells treated with colchicine alone exhibited increased translation rates (Fig. 2d). In addition, hypertrophic transcriptional signatures (*Nppa*, ratios of adult to fetal isoforms of *Tnni3/1* and *Myh6/7*), were evident in all three treatment groups (colchicine, PE + DMSO PE + colchicine) (Supplementary Fig. 2e), suggesting that microtubule destabilization does not prevent hypertrophic gene activation by PE, and that colchicine alone elicits a hypertrophic stress response in NRVMs (which was less evident in vivo, see Fig. 1d, e).

To obtain single-cell and spatially resolved insight into translation, we pulsed methionine depleted NRVMs with HPG,

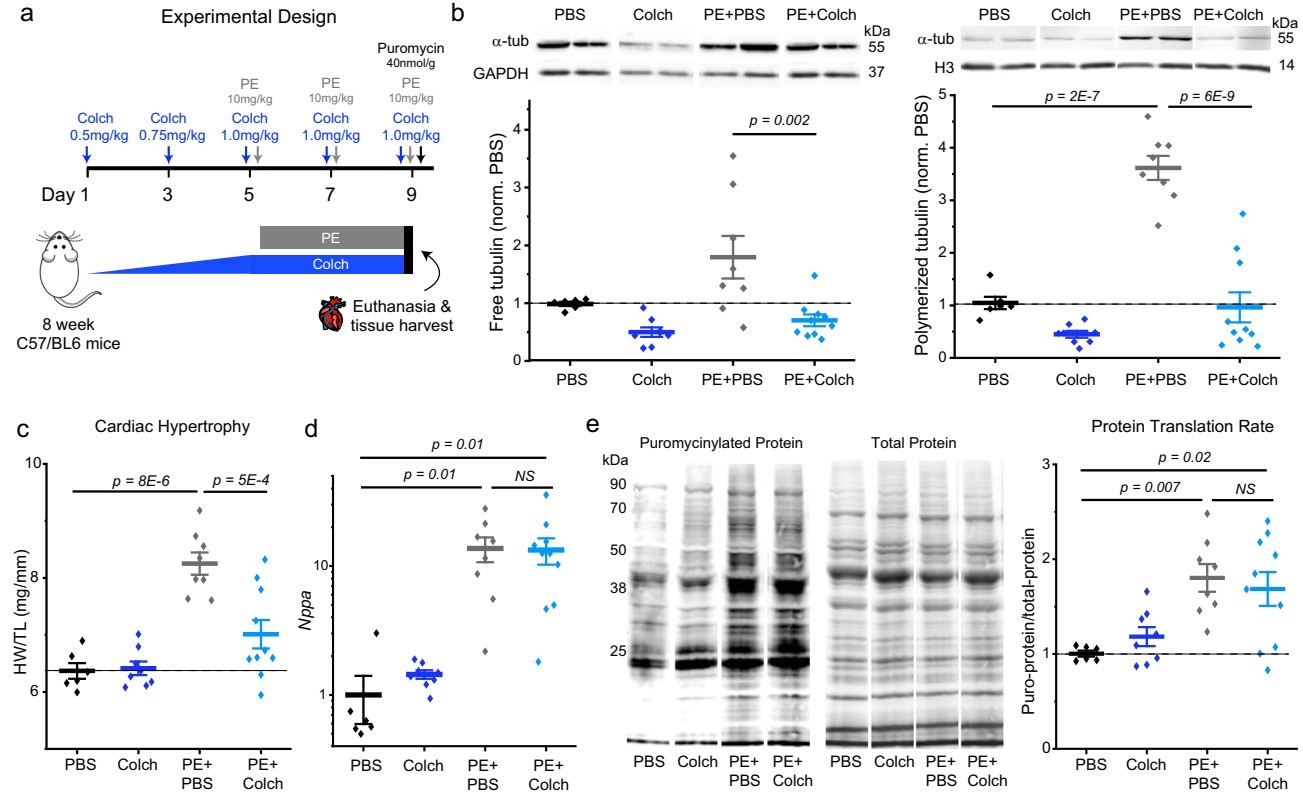

**Fig. 1 Microtubule depolymerization uncouples increased protein translation from cardiac growth. a** Experimental design for PE-induced hypertrophy mouse model. **b** Representative western blot images of free α-tubulin and loading control GAPDH (top left) and quantification of normalized free tubulin from each group (bottom left). Representative western blot images of polymerized α-tubulin and loading control Histone H3 (top right) and quantification of normalized polymerized tubulin from each group (bottom right). **c** Quantification of cardiac hypertrophy. **d** Quantification of *Nppa* expression. **e** Representative western blots for puromycinylated proteins (left) and total protein (center). Quantification of translation rate (right). For all panels, statistical significance determined via one-way ANOVA with post hoc Bonferroni comparisons. For all panels, PBS ($N = 6$), Colch ($N = 8$), PE + PBS ($N = 8$), and PE + Colch ($N = 10$). For all graphs shown in figure, the mean line is shown, with whiskers denoting standard error (SE) from the mean. Source data are provided as a Source data file.

a methionine analog that is incorporated into newly translated proteins. After washout, we used click chemistry to visualize HPG that had been incorporated into new proteins (Fig. 2e), and quantified single-cell HPG intensity as a proxy for translation rates. We found that all three treatment groups (colchicine, PE + DMSO, and PE + colchicine) exhibited significant increases in total cellular HPG intensity relative to DMSO-treated cells (Fig. 2f), with PE + colchicine exhibiting the largest increase. Taken together, we conclude that PE increases translation rates in vivo and in vitro with or without an intact microtubule network, but is unable to induce cardiomyocyte growth when microtubules are disrupted.

While global translation rates were similarly elevated with or without microtubules, the distribution of protein translation was highly skewed (Fig. 2e). PE-treated NRVMs showed a striated HPG signal with ~2 μm spacing evenly distributed throughout the cell, consistent with new protein synthesis at the sarcomere (Fig. 2e, zoom). NRVMs treated with colchicine, especially in the presence of PE, showed a substantial perinuclear accumulation of the HPG signal, more frequent cytosolic puncta, and poorly defined striations (Fig. 2e, zoom). We quantified new protein localization via 15 μm line scans from the edge of each nuclei radially out toward the end of each cell. While DMSO cells showed a linear decline in HPG fluorescence as a function of distance from the nuclear edge, PE treatment caused the nascent proteins to be distributed uniformly with distance (Fig. 2g). This indicates that during hypertrophy, there is an increase in overall

translational activity, but also a preferential increase of newly synthesized proteins distributed toward the cell ends. Colchicine-treated cells, regardless of PE, showed high perinuclear HPG fluorescence that precipitously declined within 1–2 μm, followed by a slower decline in cytosolic fluorescence with increasing distance from the nucleus. This suggests that microtubule disruption restricts a large portion of translational activity to a region proximal to the nucleus, and that microtubules are necessary for the peripheralization of new protein synthesis upon agonist stimulation.

**Microtubules localize transcripts, ribosomes, and translation.** We hypothesized that mislocalization of mRNAs could underlie the mislocalized translation upon microtubule disruption, and that microtubules may distribute mRNAs to sites of growth in response to a hypertrophic stimulus. While the precise sites of growth remain unsettled, several reports suggest that new sarcomere formation occurs at the intercalated disc (ICD) and periphery of the cardiomyocyte[15,16]. To test whether a hypertrophic stimulus alters mRNA distribution, we performed single-molecule RNA fluorescence in situ hybridization (smFISH) in NRVMs and probed for transcripts encoding proteins necessary for sarcomere (*Actc1*, alpha cardiac actin) and ICD (*Dsp*, desmoplakin) structure. *Actc1* transcripts were numerous and evenly distributed throughout the cell under control conditions, as expected given its abundance (Fig. 3a). Upon PE treatment, *Actc1* transcripts were redistributed, with a striking depletion from the

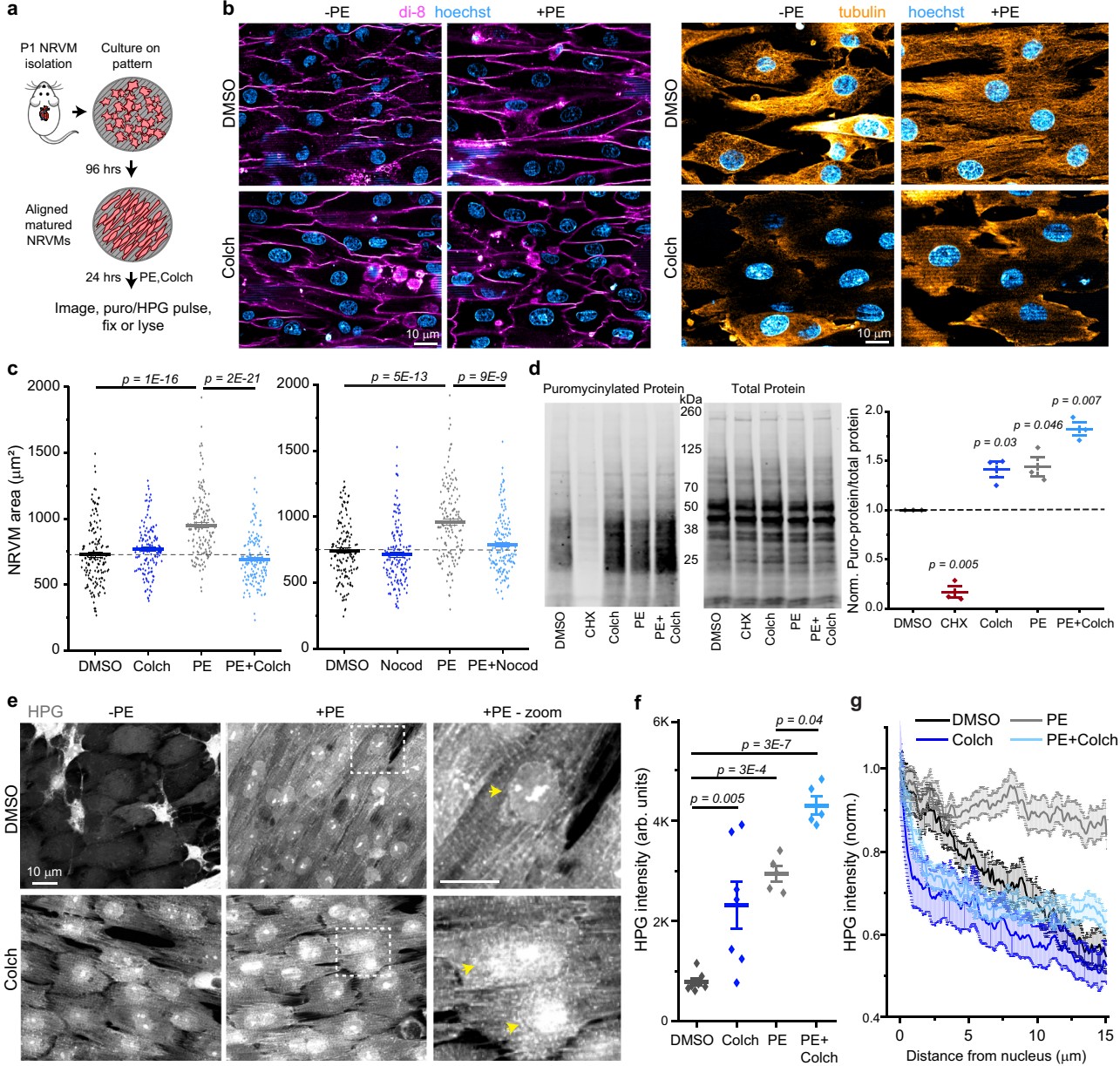

**Fig. 2 Microtubule depolymerization leads to mislocalized translation. a** Experimental design of NRVM isolation, patterning, and experiments.
**b** Representative live-cell (left) or immunofluorescence (right) images of NRVMs treated with DMSO, PE, Colch, or PE + Colch. **c** Quantification of cell areas in NRVMs treated with (left) DMSO ($n = 147$), PE ($n = 148$), Colch ($n = 138$), or PE + Colch ($n = 143$) and (right) in NRVMs treated with DMSO ($n = 145$), PE ($n = 148$), Nocod ($n = 150$), or PE + Nocod ($n = 151$). Statistical significance determined via one-way ANOVA with post hoc Bonferroni comparison, each treatment repeated in at least 3 independent NRVM litters. **d** Measurement of translational activity in NRVMs. Representative western blots labeled for puromycinylated proteins (left) and total protein (center). (right) Quantification of translation rate, normalized to DMSO. One-sample, two-tailed $t$-test vs. mean = 1, unadjusted $p$-value reported. **e** Representative fluorescence images of NRVMs treated with DMSO, PE, Colch, or PE + Colch. Yellow arrows indicate nuclei. **f** Quantification of HPG intensity. DMSO ($n = 137$ NRVMs), PE ($n = 126$), Colch ($n = 132$), or PE + Colch ($n = 140$), each data point represents pooled single-cell values from a field of view, statistical significance determined via one-way ANOVA with post hoc Bonferroni comparison. **g** Averaged line scans of HPG intensity. DMSO ($n = 30$ NRVMs), PE ($n = 30$), Colch ($n = 30$), or PE + Colch ($n = 30$). For all graphs shown in figure, the mean line is shown, with whiskers denoting standard error (SE) from the mean. Source data are provided as a Source data file.

perinuclear space and concentration to the ends of cells (Fig. 3a). To quantify the spatial distribution, the fluorescence intensity of *Actc1* was measured via line scans from the center of the nucleus to the farthest edge of each cell (Fig. 3b). PE treatment increased relative *Actc1* intensity as a function of distance, indicating a redistribution of the sarcomere transcript to the ends of cells during hypertrophy (Fig. 3b). Colchicine fully blocked the PE-induced peripheralization of *Actc1* (Fig. 3b).

*Dsp* transcripts were far less numerous than *Actc1* and formed discrete puncta in NRVMs (Fig. 3c). To quantify changes in localization, we calculated the percentage of puncta in the nucleus, the cytosol, and a 2-μm-wide periphery (Fig. 3d). In control cells, the majority of puncta resided in the cytosol, and PE treatment promoted redistribution to the periphery (Fig. 3c, d). Colchicine treatment led to a loss of *Dsp* from the periphery, accumulation around the nucleus, and prevented the PE-

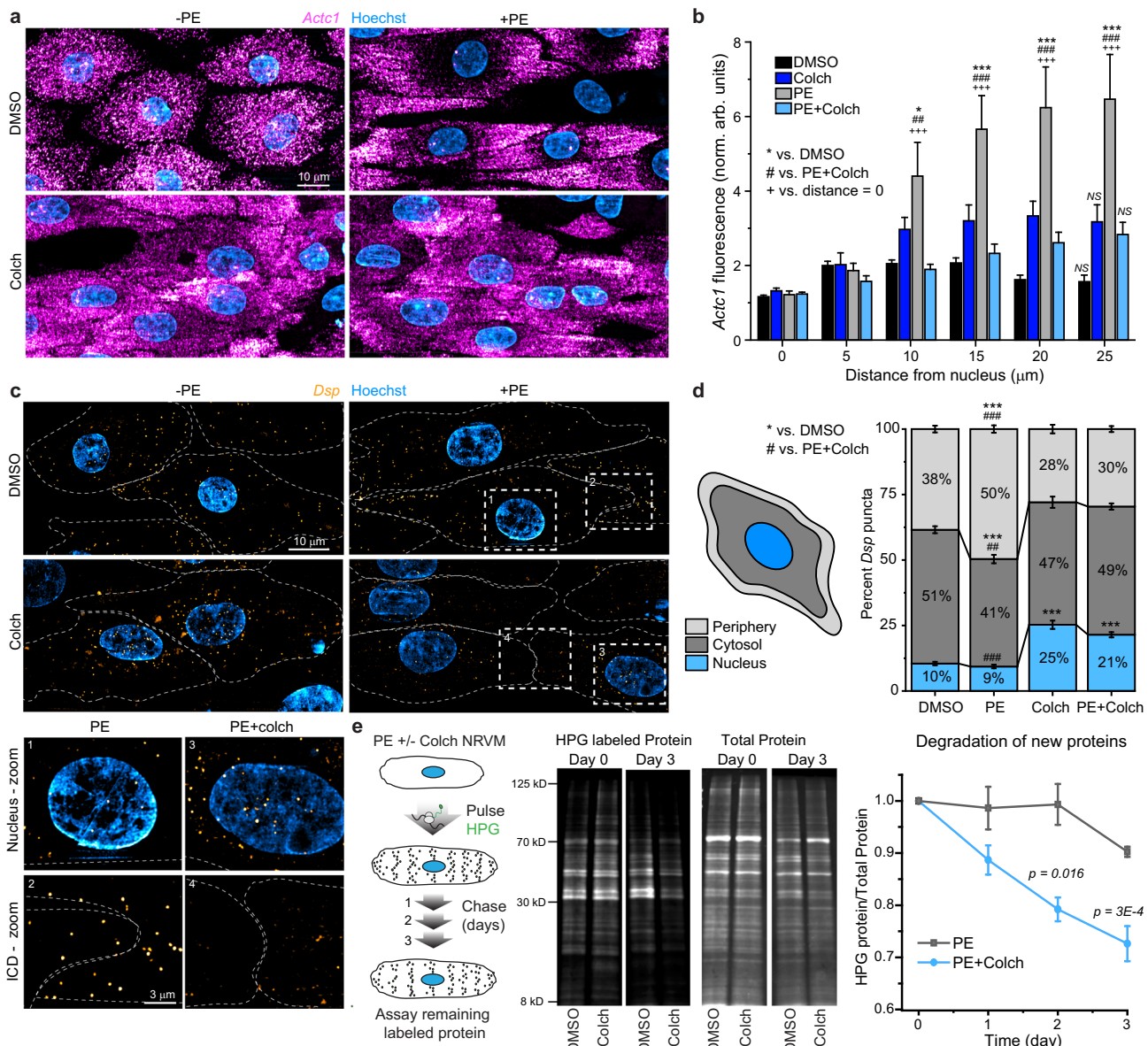

**Fig. 3 Microtubule-dependent peripheralization of mRNA transcripts is concomitant with cardiac hypertrophy. a** Representative images of NRVMs treated with DMSO, PE, Colch, and PE + Colch and hybridized for *Actc1* mRNA. **b** Quantitative line scan analysis of images from **a**. After normalization to the initial value, mean fluorescence intensities were binned every 5 μm from the center of the nucleus outward for each cell. DMSO ($n = 96$), PE ($n = 85$), Colch ($n = 82$), and PE + Colch ($n = 103$). Significance of PE group from either DMSO (*), PE + Colch (#) or initial PE measurement at distance = 0 (+) indicated by number of symbols, i.e., #$p < 0.05$, ##$p < 0.01$, and ###$p < 0.001$. Corresponding data points were omitted in overlay due to high $n$ value and high variability, causing difficulty in visualization. **c** Representative images of NRVMs treated with DMSO, PE, Colch, and PE + Colch and hybridized for *Dsp* mRNA. Numbered insets show zoomed-in regions. **d** Quantification of **c**. DMSO ($n = 82$), PE ($n = 88$), Colch ($n = 76$), and PE + Colch ($n = 87$). Significance of PE group from either DMSO (*), PE + Colch (#) indicated by number of symbols, i.e., ##$p < 0.01$ and *** or ###$p < 0.001$. **e** (left) Experimental design for measurement of protein degradation rate using pulse HPG labeling. (middle) Representative western blots of HPG-labeled protein over time. (right) Quantification of relative remaining HPG over time. For all data in figure, statistical significance determined via two-way ANOVA with post hoc Bonferroni comparison, and data are from four independent NRVM litters. For all graphs shown in figure, whiskers denote standard error (SE) from the mean. Source data are provided as a Source data file.

dependent redistribution to the edges of cells (Fig. 3c zoom, d). Taken together, these data suggest that during NRVM hypertrophy, the microtubule network directs transcripts to sites of growth and peripheralizes translation.

As noted above, PE still increased protein translation in the absence of microtubules (Figs. 1d and 2d), despite the lack of growth. This brought into question the fate of the newly synthesized, yet mislocalized proteins. As orphaned proteins may have decreased stability when not properly incorporated into

a complex such as the sarcomere[17], we hypothesized that mislocalized nascent peptides may be preferentially subject to degradation. To test this, we measured the degradation rates of newly synthesized protein by pulsing NRVMs with HPG in the presence of PE to label new proteins, washing out the excess HPG, and then assaying the amount of remaining labeled protein over 3 days (Fig. 3e). While newly synthesized proteins were largely stable over several days in PE-treated NRVMs, colchicine co-treatment significantly increased the rate of protein

degradation (Fig. 3e). These observations are consistent with mislocalized translation promoting preferential degradation, perhaps due to increased access to the protein clearance machinery without proper incorporation into a multimeric complex such as the sarcomere.

**Microtubules orchestrate translation in vivo**. While NRVMs are a useful system to study hypertrophy, they lack the rigid cytoarchitecture of fully mature cardiomyocytes. To more carefully interrogate the localization of mRNAs and translation, we used isolated adult rat ventricular cardiomyocytes (ARVMs), which are stereotypically structured and do not change shape during short-term culture or upon microtubule disruption[13]. This allows the assessment of bona fide subcellular redistribution independent of changing cell morphology.

We first pulsed ARVMs with HPG to assess localization of newly translated proteins in the presence or absence of microtubules. In DMSO-treated cells, HPG showed an evenly distributed, striated pattern consistent with Z-disc spacing, and a clear enrichment of signal at the ICD (Fig. 4a, top left). Colchicine treatment led to a collapse of the HPG signal around the nucleus that decayed with distance, indicating a diffusion restricted pattern, and abolishment of ICD enrichment (Fig. 4a, bottom left). To quantify nuclear accumulation, we calculated the ratio of HPG fluorescence in perinuclear to cytosolic ROIs (perinuclear enrichment), and ICD enrichment was assessed by the ICD/C ratio (Fig. 4c, cartoon). Colchicine increased perinuclear enrichment and eliminated ICD enrichment, indicating that microtubule destabilization grossly mislocalizes newly synthesized proteins in adult cardiomyocytes (Fig. 4b, left).

We next tested if new peptide mislocalization is attributable to a mislocalization of ribosomes, which were labeled using smFISH with probes against 18S ribosomal RNA (rRNA). Similar to HPG localization, we observed a striated ribosomal pattern and enrichment at the ICD in control cells. Microtubule disruption led to drastic confinement of ribosomes to the perinuclear space (Fig. 4a and b, middle). To control for colchicine-specific effects, we repeated this experiment using nocodazole, first verifying that high dose nocodazole (10 μM) is sufficient to completely destabilize the microtubule network (Supplementary Fig. 4a). Nocodazole treatment caused a similar collapse of ribosomes around the nucleus, corroborating that the observed phenotype is due to microtubule destabilization (Supplementary Fig. 4a). In addition, we confirmed these findings with a secondary method using immunofluorescence against Rps6, a subunit within the 40S ribosome, which phenocopied these results (Supplementary Fig. 4b). This data indicates that an intact microtubule network is indispensable for subcellular positioning of ribosomes in mature cardiomyocytes.

To extend our observations to human cardiomyocytes, we isolated left ventricular myocytes from a non-failing human heart and visualized ribosome localization after 16 h colchicine treatment. As observed in ARVMs, ribosomes in human cardiomyocytes showed a striated distribution throughout the cell and robust enrichment at the ICD; microtubule destabilization with colchicine led to a depletion of ribosomes from the ICD and collapse around the nucleus (Supplementary Fig. 4c).

Finally, we assessed if mRNA localization is similarly controlled by microtubules. Using smFISH, we probed all polyadenylated (polyA) mRNA in the cell using a 30-mer dT fluorescently labeled oligo. polyA mRNA localization mirrored that of mature ribosomes, showing a striated pattern with ICD enrichment, suggesting that mRNA is transported from the nucleus and locally translated in the cardiomyocyte. Again, both colchicine and nocodazole treatment caused a robust perinuclear

collapse of mRNAs with a loss of signal at the ICD (Fig. 4a right, Fig. 4b right, Supplementary Fig. 4a). This phenotypic collapse in the distribution of mRNAs and ribosomes was dramatic and observed in the majority of cells (Supplementary Fig. 4d).

We next evaluated the time course of mRNA redistribution upon microtubule disruption. Three hours of colchicine treatment was required to detect a decrease in the mRNA signal at the ICD and accumulation around the nucleus, with further loss of cytosolic signal and nuclear accumulation occurring gradually over 24 h (Fig. 4c). As microtubules are almost completely depolymerized within 1 h under these conditions[13], these data are consistent with the half-lives of most transcripts exceeding that duration, and suggest that once localized, mRNAs can remain distributed in the absence of microtubules, but new transcripts are confined to the nuclear proximity and do not replace distal stores.

We also probed specific transcripts of Actc1, Dsp, and Myh6 (cardiac alpha myosin heavy chain, a second essential sarcomeric component) (Supplementary Fig. 5). In control cells, both Actc1 and Myh6 puncta were abundant, evenly distributed throughout the entire cell and often enriched at sarcomeres (Supplementary Fig. 5a, b). Colchicine treatment led to a sharp increase in transcript abundance around the nuclei for both transcripts, with decreased signal between the two nuclei as well as progressive drop in signal toward the ends of the cell (Supplementary Fig. 5a, b). For the less abundant Dsp puncta, there was a strong twofold enrichment of transcripts at the ICD (Supplementary Fig. 5c), consistent with a recent report indicating that specific transcripts localize to sites of respective protein function[4]. Colchicine treatment increased puncta accumulation around the nucleus and abolished enrichment at the ICD (Supplementary Fig. 5d). Together, this data suggests that microtubules control subcellular localization of diverse mRNAs in the cardiomyocyte.

In non-muscle cells, well-studied mRNAs (β-actin, ASH1, oskar) have been shown to use the actin and microtubule cytoskeleton for distribution, with some cell types utilizing microtubules for long-range transport and actin for local trafficking[18]. To determine if the actomyosin network could represent a parallel or additional pathway for mRNA localization, we inhibited myosin motors using either blebbistatin or Y27632 treatment or inhibited actin polymerization with Latrunculin A and quantified the localization of mRNA and new protein synthesis (Fig. 4d). None of these treatments promoted nuclear accumulation of mRNA or newly synthesized protein compared to control cells. Latrunculin A showed a minor disruption of ICD enrichment for both polyA mRNA and HPG, but this was not recapitulated by the other treatments (Fig. 4e). Overall, these data indicate that the majority of mRNA localization, and consequently, localization of protein synthesis, in the adult cardiomyocyte is controlled by microtubules, with minimal or no dependence on actinomyosin-based transport.

We next sought to confirm the role of microtubules in positioning the translational machinery in vivo. A single 1 mg/kg colchicine injection is well-tolerated in rodents[19], and likely results in a concentration of ~1–2 μM colchicine in myocardium that peaks around 20 min and declines over time (extrapolated from ref. [20]). As such, we injected rats with 1 mg/kg colchicine and harvested cells and tissue 16–20 h later. Isolated cardiomyocytes were immediately fixed and hybridized for polyA mRNA and antibody-labeled for microtubules. Though we observed less microtubule depolymerization than with in vitro treatment, in vivo colchicine administration was sufficient to disrupt microtubules and cause perinuclear accumulation of mRNA with a loss of transcripts at the edges of the cells (Supplementary Fig. 4e).

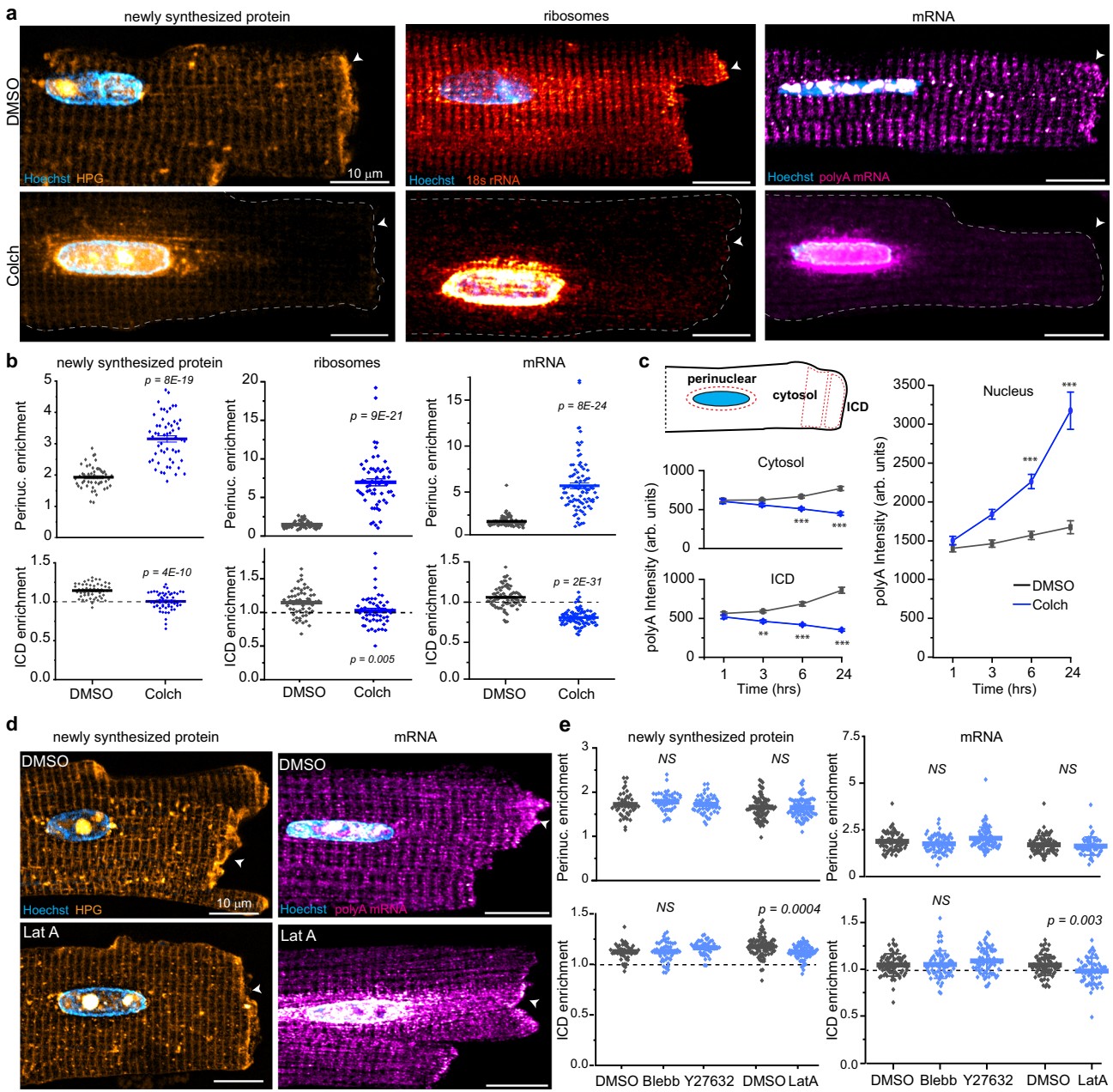

**Fig. 4 Microtubules control the localization of mRNA, ribosomes, and new protein synthesis. a** Representative images of (left) newly synthesized protein, (middle) ribosomes and (right) mRNA in (top) DMSO or (bottom) Colch-treated ARVMs. White arrow indicates ICD. **b** Nuclear:cytosolic and ICD:cytosolic ratios of mean fluorescence intensities of experiments shown in **a**. Newly synthesized protein: DMSO ($N = 3$, $n = 52$), Colch ($N = 3$, $n = 55$); ribosomes: DMSO ($N = 2$, $n = 54$), Colch ($N = 2$, $n = 57$; mRNA: DMSO ($N = 3$, $n = 82$), Colch ($N = 3$, $n = 90$). Statistical significance determined via two sample, two-tailed $t$-test. **c** Cartoon (left, top) depicting representative placement of perinuclear, cytosolic, and ICD ROIs for analysis in **b** and **c**. Mean polyA fluorescence intensity in the cytosol (left, middle), ICD (left, bottom), and nucleus (right) as a function of time. DMSO, $n = 47$ (1 h), $n = 51$ (3 h), $n = 54$ (6 h), $n = 40$ (24 h). Colch, $n = 49$ (1 h), $n = 47$ (3 h), $n = 50$ (6 h), $n = 47$ (24 h). Statistical significance determined via one-way ANOVA with post hoc Bonferroni comparison. **d** Representative images of (left) newly synthesized protein and (right) mRNA in (top) DMSO or (bottom) Latrunculin A-treated ARVMs. White arrow indicates ICD. **e** Nuclear:cytosolic and ICD:cytosolic ratios of mean fluorescence intensities of experiments shown in **d**. Newly synthesized protein: DMSO ($N = 2$, $n = 46$), Blebb ($N = 2$, $n = 49$), Y27632 ($N = 2$, $n = 48$), DMSO ($N = 3$, $n = 82$), LatA ($N = 3$, $n = 80$); mRNA: DMSO ($N = 3$, $n = 70$), Blebb ($N = 3$, $n = 63$), Y27632 ($N = 3$, $n = 74$), and DMSO ($N = 3$, $n = 77$), LatA ($N = 3$, $n = 59$). Statistical significance determined via one-way ANOVA with post hoc Bonferroni comparision (DMSO, Blebb, Y27632) and two sample, two-tailed $t$-test (DMSO, LatA). For all graphs shown in figure, the mean line is shown, with whiskers denoting standard error (SE) from the mean. Source data are provided as a Source data file.

We next sought to assess mRNA and ribosome localization in intact tissue. We first probed for polyA mRNA in ventricular tissue sections from PBS or colchicine-injected rats. While colchicine-treated myocardium showed a lack of striated polyA signal and accumulation around the nucleus, staining was often weak or uneven and unsuitable for quantification (Supplementary Fig. 6b). However, ribosomal labeling with 18S rRNA was uniform and exhibited excellent signal to noise in tissue sections. To confirm tissue quality and mark locations of interest, we performed simultaneous immunofluorescence against desmin, an

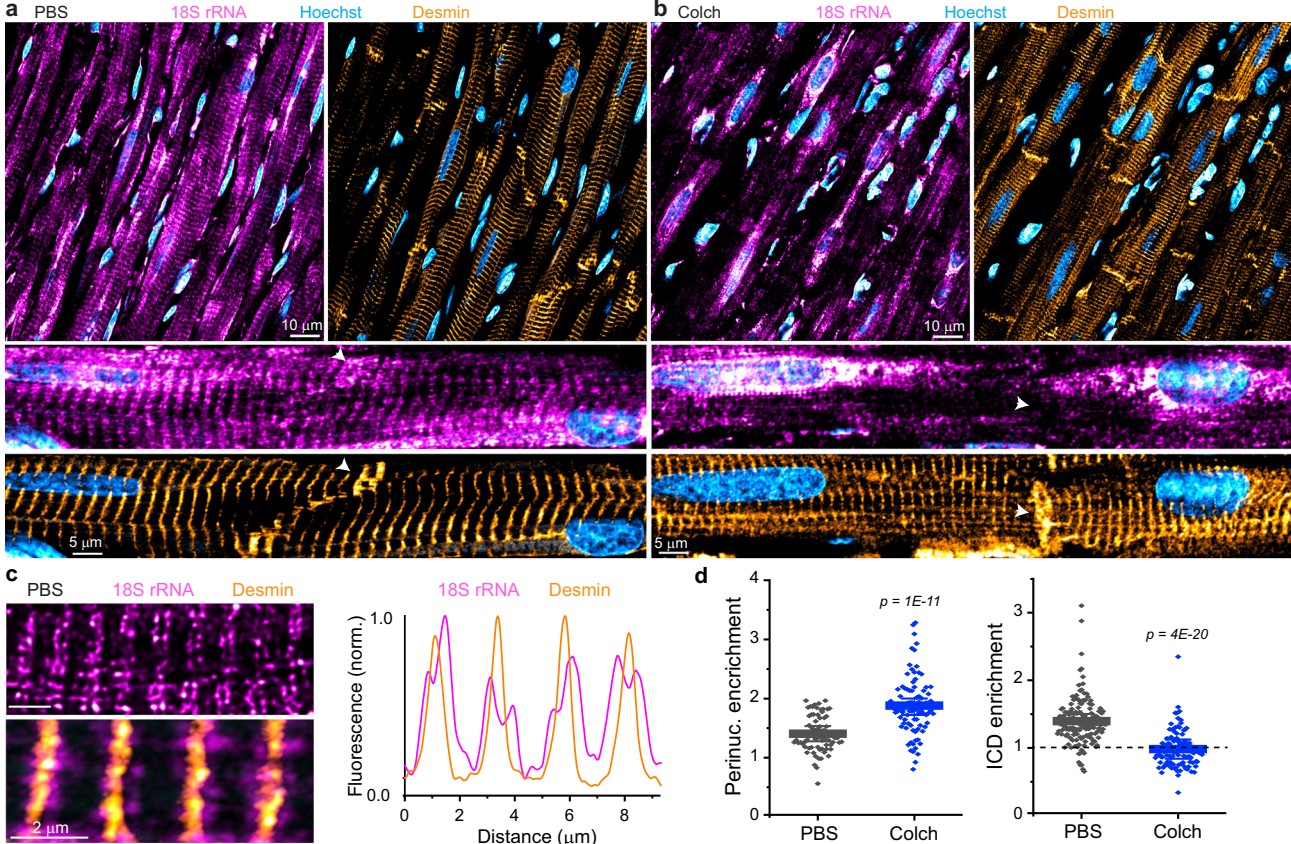

**Fig. 5 Microtubules control the localization of ribosomes in vivo. a** Representative image of PBS-injected rat ventricular myocardium, with zoomed-in image below and white arrow highlighting ICD region. **b** Representative image of Colch-injected rat ventricular myocardium with similar zoom and highlight as in **a**. **c** (left) Zoomed-in image of PBS-injected rat ventricular myocardium. (right) Line scan analysis of the composite image shown at bottom left. **d** Nuclear:cytosolic and ICD:cytosolic ratios of 18S rRNA mean fluorescence intensities of experiment depicted in **a** and **b**. PBS (N = 2), (n = 71, N/C), (n = 137, ICD/C), Colch (N = 2), (n = 92, N/C), (n = 108, ICD/C). Statistical significance determined via two sample, two-tailed t-test. For all graphs shown in figure, the mean line is shown, with whiskers denoting standard error (SE) from the mean. Source data are provided as a Source data file.

intermediate filament that localizes to the Z-disc and ICD (Fig. 5a, b). Control tissue sections showed an evenly distributed sarcomeric banding pattern of 18s rRNA with consistent enrichment in regions corresponding to the ICD (Fig. 5a), consistent with in vitro findings. At high magnification, ribosomes can be observed to localize to either side of each Z-disc, forming a repeating pattern of doublets within the cell (Fig. 5c). Colchicine-injected rats showed tissue-wide disruption of this striated distribution of ribosomes, increased ribosomal labeling around cardiomyocyte nuclei, and a marked and consistent absence of ribosomes at cell–cell junctions, despite normal desmin labeling (Fig. 5b). Using a similar ratiometric approach to quantify these differences (Supplementary Fig. 6a), we found that a single colchicine injection was sufficient to cause a perinuclear collapse of ribosomes and abolish enrichment at the ICD (Fig. 5d), indicating that microtubules control the positioning of the translational machinery in vivo.

**Kinesin-1 transports cardiomyocyte mRNAs.** The microtubule motors kinesin and dynein regulate organelle transport and intracellular trafficking, and are known to play a role in the transport of mRNA in non-muscle cells[21,22]. As Kinesin-1, encoded by *Kif5b*, is the most highly expressed kinesin in ARVMs, we hypothesized that Kinesin-1 may actively transport mRNA in the cardiomyocyte. To test this, we generated adenovirus encoding shRNA to knockdown *Kif5b* in ARVMs. Viral expression resulted in a time-dependent knockdown of *Kif5b*, which required 96 h to achieve ~70%

knockdown (Fig. 6a), consistent with the long half-life of Kinesin-1 protein. Kinesin-1 knockdown caused no gross morphological changes to adult cardiomyocytes after 96 h (Supplementary Fig. 7), nor did it overtly alter the organization of the microtubule network (Fig. 6b). However, *Kif5b* knockdown dramatically mislocalized polyA mRNA and led to a nuclear accumulation of ribosomes (Fig. 6c). These data indicate that Kinesin-1 is required for the transport and spatial distribution of mRNA and ribosomes in cardiomyocytes.

As our overall model proposes that the proper localization of mRNAs and translation is required for cardiac hypertrophy, we thus hypothesized that Kinesin-1 depletion would be sufficient to prevent growth in NRVMs. To test this hypothesis, we examined PE-induced growth after knockdown of *Kif5b* in NRVMs. Viral expression of *Kif5b* shRNA resulted in ~70% knockdown after 72 h in NRVMs (Fig. 6d). We then treated *Kif5b* KD cells or cells expressing a scramble virus with PE to assess their ability to grow. As expected, PE induced a robust cell size increase in scramble NRVMs after 24 h of treatment, while PE-induced hypertrophy was blocked by *Kif5b* knockdown (Fig. 6e, f). Together, these data indicate that Kinesin-1-mediated transport of mRNA and the translational machinery is necessary for growth of the cardiomyocyte.

**Discussion**
Here we identify an essential role of microtubule motor-dependent transport of mRNAs and ribosomes for the

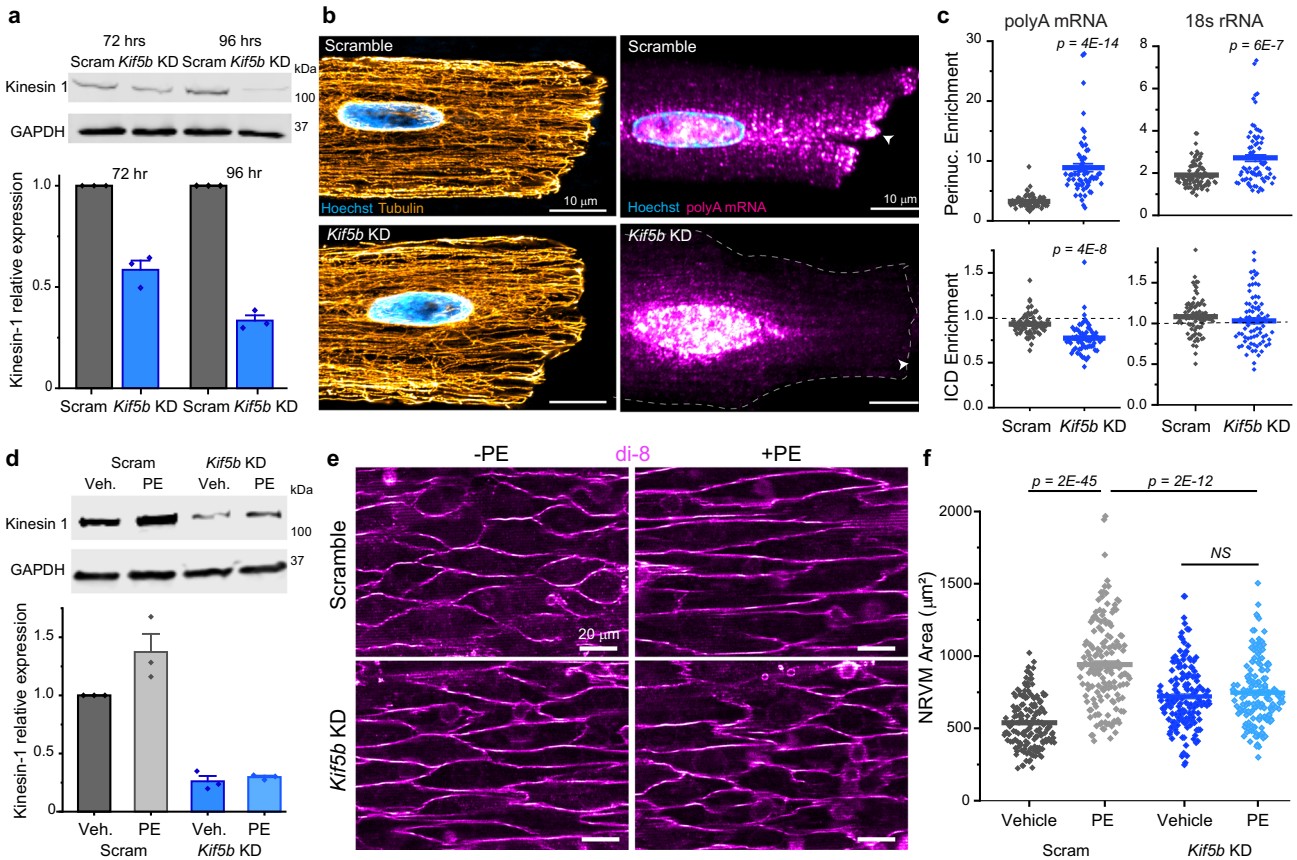

**Fig. 6 mRNA localization and cardiac hypertrophy depends on the microtubule motor Kinesin-1. a** (top) Representative western blot images of kinesin-1 and loading control GADPH in ARVMs. (bottom) Quantification of relative kinesin-1 expression, normalized to GAPDH. $N = 3$. **b** Representative immunofluorescence images of the microtubule network (left) and polyA mRNA (right) in Scramble and *Kif5b* KD cells. White arrow indicates ICD. **c** Nuclear:cytosolic and ICD:cytosolic ratios of polyA mean fluorescence (left) and 18s rRNA mean fluorescence (right) in Scramble and *Kif5b* KD cells. Scramble ($N = 2$, $n = 62$, polyA) ($N = 2$, $n = 88$, 18s), *Kif5b* KD ($N = 2$, $n = 61$, polyA) ($n = 2$, $N = 76$, 18s). Statistical significance determined via two sample, two-tailed *t*-test. **d** (top) Representative western blot images of kinesin-1 and loading control GADPH in NRVMs. (bottom) Quantification of relative kinesin-1 expression, normalized to GAPDH. $N = 3$. **e** Representative live-cell images of (top row) Scramble or (bottom row) *Kif5b* KD NRVMS, treated with (left column) vehicle or (right column) PE. **f** Quantification of cell areas in NRVMs from experiment shown in **e**. Scramble+vehicle ($N = 3$, $n = 160$), Scramble+PE ($N = 3$, $n = 160$), *Kif5b* KD + vehicle ($N = 3$, $n = 160$), *Kif5b* KD + PE ($N = 3$, $n = 160$). For all graphs shown in figure, the mean line is shown, with whiskers denoting standard error (SE) from the mean. Source data are provided as a Source data file.

spatiotemporal control of translation in cardiomyocytes. Our data indicate that translation location, and not simply a global increase in translation per se, is a critical determinant of cardiac hypertrophy. Upon cardiac stress, microtubules effectively couple an increase in transcription and translation to productive growth.

These findings have implications beyond hypertrophic growth. We find microtubules are indispensable for the homeostatic distribution of mRNAs, ribosomes, and translation in heart muscle cells, indicating an essential role in protein quality control. Sarcomeres are replaced weekly while cardiomyocytes have a lifespan of years[23], underscoring the need for such mechanisms. Further, our work posits an evolutionary advantage for localizing RNAs to sites of protein function, both to support directed growth as well as to promote nascent peptide stability and protection from degradation.

Our findings support a model of local translation, whereby transcripts are trafficked to and translated at sites of protein function. Such a mechanism may confer numerous advantages in a muscle cell. Active trafficking may more efficiently localize translation compared to diffusional mechanisms in densely packed and polarized cells[24,25], and local translation can ultimately reduce transport cost by translating multiple proteins from one transcript. Local translation can facilitate multi-protein

complex assemblies (such as sarcomeres or ICDs) through high local concentrations[26–29] and by ensuring that nascent proteins can fold properly into a pre-existing cytoskeleton[30,31], while also restricting a protein's function in space and time. Pre-existing scaffolds or associated chaperones may serve as quality control mechanisms to prevent nascent peptides from misfolding, consistent with mislocalized translation accelerating degradation of nascent peptides (Fig. 2g).

The precise locations of new sarcomere addition remain to be determined. We observe microtubule-dependent relocation of sarcomeric transcripts and nascent peptides to the ends of cells upon an eccentric growth stimulus, complementing previous reports suggesting new sarcomere addition at the ICD[15,16], which we find particularly enriched with transcripts and translational machinery. The mechanisms that guide motor-dependent peripheralization of sarcomeric building blocks, and how these may be differently utilized in different forms of directional remodeling (i.e., concentric vs. eccentric hypertrophy), represents a key area for future study.

Numerous mechanisms may dynamically tune microtubule-dependent trafficking to match changing cardiac demand. The density and directionality of microtubule tracks can be quickly remodeled via microtubule growth and shrinkage dynamics. The

affinity and processivity of motor proteins are highly dependent upon post-translational modifications to the motors, to structural microtubule-associated proteins (MAPs) that bind microtubule tracks, as well as to the tracks themselves. For example, post-translational modification of MAP4, the most highly expressed cardiac MAP, is sufficient to alter both microtubule stability and mRNP motility[32,33], and detyrosination of α-tubulin also alters Kinesin-1-based trafficking in non-muscle cells[34,35]. Both MAP4 and detyrosination are induced in cardiac hypertrophy and heart failure in patients[7,32], implying a potentially causal role in cardiac remodeling. How these cytoskeletal changes are integrated during cardiac remodeling, and if the timing of such changes contributes to the progression from hypertrophy to heart failure remains to be determined. Early microtubule proliferation may aid mRNA transport to facilitate growth, whereas profound cytoskeletal remodeling may be linked to the global disruption of protein homeostasis and proteotoxicity often seen in the failing heart. Further research is required to understand how the control of mRNA and ribosome localization contributes to cardiovascular disease progression.

## Methods

**Animals.** Animal care and use procedures were performed in accordance with the standards set forth by the University of Pennsylvania Institutional Animal Care and Use Committee and the Guide for the Care and Use of Laboratory Animals published by the US National Institutes of Health; protocols were approved by the University of Pennsylvania Institutional Animal Care and Use Committee. Both rats and mice were housed in a facility with 12-h light/dark cycles and provided ad libitum access to water and chow. Temperature and humidity were checked daily to ensure that these parameters stay within appropriate ranges (20–26 °C, 30–70%, respectively).

**Adult rat cardiomyocyte (ARVM) isolation.** Primary adult ventricular myocytes were isolated from 8- to 12-week-old Sprague Dawley rats using Langendorff retrograde aortic perfusion with an enzymatic solution. Briefly, the heart was removed from an anesthetized rat under isoflurane and retrograde-perfused on a Langendorff apparatus with a collagenase solution. The digested heart was then minced and triturated with glass pipettes to free individual cardiomyocytes. The resulting supernatant was separated and centrifuged at 300 rpm to isolate cardiomyocytes. These cardiomyocytes were then resuspended in rat cardiomyocyte media (Medium 199 (Thermo Fisher) supplemented with 1x insulin-transferrin-selenium-X (Gibco), 1 μg/μL primocin (InvivoGen), and 20 mM HEPES, pH = 7.4 (UPenn Cell Center)) at low density, cultured at 37 °C and 5% $CO_2$ with the addition of 25 μmol/L of cytochalasin D in the media. Cells were excluded from experiments/analysis if they were hypercontracted or displayed loss of membrane integrity.

**Neonatal rat ventricular myocyte (NRVM) isolation.** NRVMs were isolated from 1- to 2-day-old litters. Pups were anesthetized on ice, rapidly decapitated and their hearts were excised and placed in chilled Hanks' Balanced Salt Solution (HBSS, Sigma-Aldrich). Pooled hearts from a single litter were minced and digested in HBSS containing trypsin (Worthington Biochemical) and benzonase (Sigma-Aldrich). Cells were gently centrifuged, resuspended in serum-containing NRVM media (DMEM (Thermo Fisher), 5% FBS, 12.5 mM HEPES (UPenn Cell Center), 4 mM Aln-Gln (Sigma-Aldrich), 0.1 mg/ml primocin (InvivoGen)), and pre-plated for 2 h in cell-culture treated flasks to remove the majority of cardiac fibroblasts. NRVMs were then plated onto nano-patterned culture dishes (Curi Bio) at a seeding density of 150,000 cells per $cm^2$. After overnight attachment in serum-containing media, the cells were serum starved for 48 h before treatments.

**Human myocardial tissue and isolation.** Procurement of human myocardial tissue was performed under protocols and ethical regulations approved by Institutional Review Boards at the University of Pennsylvania and the Gift-of-Life Donor Program (Pennsylvania, USA). Informed consent was obtained from subjects or relatives. Human studies were conducted in compliance with the principles of the Declaration of Helsinki. The non-failing heart was obtained at the time of organ donation from a cadaveric donor. Hearts received cold, blood-containing, high-potassium cardioplegic solution in vivo. Explanted hearts were transported from the operating suite to the laboratory in cold Krebs–Henseleit buffer (KHB) solution (12.5 mM glucose, 5.4 mM KCl, 1 mM lactic acid, 1.2 mM $MgSO_4$, 130 mM NaCl, 1.2 mM $NaH_2PO_4$, 25 mM $NaHCO_3$, and 2 mM Na pyruvate, pH 7.4). Myocytes were disaggregated. Briefly, hearts were weighed and rinsed in KHB. A non-infarcted free wall region of the left ventricular apex was dissected and a small catheter was placed into the lumen of the left ventricular descending artery. Major large vessels on the tissue piece were identified by

injecting KHB via the cannula and tied by suture knots to improve perfusion via small vessels. Once the tissue was ready for perfusion, it was covered by plastic wrap with pores for outflow, in order to maintain tissue temperature at 37 °C. The cannulated left ventricular tissue was perfused with a non-recirculating $Ca^{2+}$-free solution (KHB containing 20 mM BDM and 10 mM taurine) for 10–15 min until the outflow temperature reached around 37 °C. Then, 200 ml of KHB containing 294 units $ml^{-1}$ collagenase, 20 mM BDM, and 10 mM taurine was perfused for 3 min without recirculation followed by ~25 min. $Ca^{2+}$ was introduced stepwise per minute by adding $CaCl_2$ solution up to 1 mM (i.e., $4 \times 50$ μM, $4 \times 100$ μM, and $2 \times 200$ μM) into the recirculated collagenase solution. Then the tissue was perfused for 5 min with rinse solution (KHB containing 10 mM taurine, 20 mM BDM, 1 mM $CaCl_2$, and 1% BSA). The tissue was then removed from the cannula, and myocardial tissue was minced in the rinse solution and triturated using glass pipets. The resulting cell suspension was filtered through 280-μm nylon mesh (component supply U-CMN-280), centrifuged ($25 \times g$ for 2 min), and resuspended in rinse solution. The temperature was maintained at 37 °C throughout the isolation. Viable cells were enriched by gravity sedimentation for 5 min, and the resulting loose pellet was transferred to a fresh tube and resuspended in the proper amount of normal Tyrode's solution.

**In vivo colchicine and phenylephrine (PE) treatment.** For chronic colchicine treatment of mice, 8–12-week-old male C57BL/6 wild-type mice were randomly assigned to vehicle, colchicine, PE, and colchicine/PE treatment groups. Colchicine treatment followed a modified protocol from refs. [10] and [36]. Briefly, mice were I.P. injected every other day with colchicine in increasing doses (0.5, 0.75, and 1 mg/kg body weight) to improve tolerance[9] and were maintained at the 1 mg/kg BW dose. Control animals were I.P. injected with PBS. PE (10 mg/kg BW in 0.2% ascorbic acid (Sigma-Aldrich) dissolved in PBS) was administered subcutaneously ~0.5–1 h after colchicine treatment (to ensure sufficient colchicine accumulation in the myocardium at the time of PE action[20]) for 3 total injections over 5 days. Control animals received subcutaneous injections of vehicle (0.2% ascorbic acid in PBS). Mice showed no overt signs of distress or toxicity during this protocol, and on day 9 were euthanized for tissue harvesting. For acute colchicine treatment of rats, rats were injected I.P. with 1 mg/kg body weight sterile colchicine (Sigma-Aldrich) dissolved in PBS. The animals were euthanized 16–20 h after injection with colchicine.

**In vivo puromycinylation.** Mice were injected I.P. with puromycin (40 nmol/g BW diluted in PBS). After $30 \pm 1$ min, animals were euthanized, and the hearts were extracted in ice-cold PBS.

**In vitro pharmaceuticals.** Colchicine (10 μM in DMSO, Sigma-Aldrich), Phenylephrine (100 μM in culture media, Sigma-Aldrich), Isoproterenol (1 μM in DMSO), Nocodazole (10 μM in DMSO, Thermo Fisher), Puromycin (10 μg/mL solution, A.G. Scientific), Cycloheximide (20 μg/mL in DMSO, Sigma-Aldrich), Blebbistatin (5 μM in DMSO, Cayman Chemical), Latrunculin A (10 μM in DMSO, Abcam), Y27632 (10 μM in DMSO, Sigma-Aldrich). Cells were treated with colchicine, nocodazole, or DMSO for 3 h prior to treatment with PE.

**Virus generation and vectors.** The adenovirus encoding shKif5b construct was directed toward a single target site under the U6 promoter. It was generated and produced by inserting appropriate cDNAs into pENTR for further Gateway recombination in adenoviral expression plasmids. Constructs were then transferred by Gateway recombinase into adenoviral expression plasmid pAdCMV/V5/DEST (Invitrogen). Recombinant adenoviral vectors were produced and amplified in HEK 293a cells according to manufacturer protocol (ViraPower Adenoviral Expression System; Invitrogen). Viruses were collected by CsCl gradient centrifugation and dialyzed against a 5% sucrose buffer as a final step. Target site for shKif5b: gcacacagactgagagcaaca. eBFP2 (enhanced blue variant of green fluorescent protein) was used as a marker of transduction for shKif5b. The adenovirus encoding shScramble (no fluorophore) was purchased from Vector Biolabs.

**NRVM cell size measurement.** After 24 h treatment with Colchicine and/or PE, NRVMs were stained with Di-8ANEPPS and Hoechst for 10 min at 37 °C to label the cell membrane and nuclei, respectively. Similarly, NRVMs were stained with Di-8ANEPPS and Hoechst after 48 h treatment with colchicine and/or isoproterenol (1 μM) to measure cell area.

**shScram and shKif5b knockdown.** For knockdown of *Kif5b* in ARVMs, cells were treated with either 0.1 μl/ml shScram or 0.1 μl/ml shKif5b immediately after isolation. At either 72 or 96 h after virus treatment, cells were then split for either lysis preparation for western blot analysis to calculate percent knockdown, or fixed onto coverslips for smFISH. Thus, in Fig. 5, relative knockdown of *Kif5b* expression was measured (shown in Fig. 5c) for each experiment shown in Fig. 5e. For knockdown of *Kif5b* in NRVMs, cells were treated with either 0.1 μl/ml shScram or 0.1 μl/ml shKif5b at the time of serum starvation, 24 h after initial plating. At 48 h after virus treatment, NRVMs were treated with 100 μM PE for 24 h. At 72 h after virus treatment, live cells were imaged after application of DI-8-ANEPPS. Immediately after imaging, NRVMs were washed once with PBS and lysed in 1X RIPA lysis

buffer (Cayman Chemical) + 1X Protease/Phosphatase Inhibitor Cocktail (Cell Signaling) for western blot lysate preparation to calculate percent knockdown. Thus, in Fig. 6, relative knockdown of *Kif5b* expression was measured (shown in Fig. 6d) for each experiment shown in Fig. 6e, f.

**Preparation of tissue sections.** Immediately after isolation, whole hearts were perfused with RNase-free PBS to flush out blood (~1 min) and then 4% paraformaldehyde (Electron Microscopy Sciences) in RNase-free PBS until the heart was stiffened to the extent that it could no longer perfuse (~15–20 min). The heart was then left in 50 mL 4% paraformaldehyde in RNase-free PBS in a conical tube rotating at 4 ˚C overnight. Next, at room temperature, the heart was washed three times for 30 min in RNase-free PBS and then placed in 50 mL 15% sucrose solution (Thermo Fisher) rotating at 4 ˚C overnight, or until the heart no longer floated. Finally, the heart was placed in 50 mL 30% sucrose solution rotating at 4 ˚C overnight, or until the heart no longer floated. Hearts were then embedded in tissue freezing medium (Electron Microscopy Sciences) in plastic mold using liquid nitrogen-cooled isopentane prior to sectioning. Frozen blocks were sectioned to a thickness of ≤10 μm and stored at −80 ˚C.

**Fractionation.** Tissue from the left ventricle was dissected, weighed, and flash-frozen in liquid nitrogen. These tissue samples were placed into 50 μl/mg of room-temperature microtubule-stabilizing buffer (adapted from ref. [37]) containing: glycerol (50%, v/v), DMSO (5%, v/v), sodium phosphate (10 mM), $MgCl_2$ (0.5 mM), and EGTA (0.5 mM); pH 6.95. Fresh aliquots of GTP (0.5 mM, Sigma-Aldrich) and phosphatase/protease inhibitor (Cell Signaling, 1X) were added to the buffer immediately before use. Tissue samples were dounce homogenized thoroughly and centrifuged at $21,000 \times g$. The supernatant was collected as the "free" tubulin fraction and the remaining pellet was solubilized in 50 μL per mg of original tissue weight of 1X RIPA lysis buffer (Cayman Chemical) + 1X Protease/Phosphatase Inhibitor Cocktail (Cell Signaling) and 2% SDS. This "polymerized" microtubule fraction was disrupted on ice with vigorous pipetting for 1 h.

**Immunofluorescence.** Cells were fixed in 4% PFA (Electron Microscopy Sciences) for 10 min, washed three times with PBS, and permeabilized in 0.1% TritonX-100 for 10 min at room temperature. After washing twice with PBS, cells were placed in blocking buffer (1:1 Seablock (Abcam) and 0.1% TritonX-100 (Bio-Rad) in PBS) for at least 1 h at room temperature, then labeled with primary antibodies (see below) for 24–48 h at 4 ℃. Cells were then washed three times in TBS, then labeled with secondary antibodies (see below) in TBS at room temperature for 2–4 h and washed a final two times with TBS. Stained cells were mounted on #1.5 coverslips in Prolong Diamond Antifade Mountant (Thermo Fisher) for imaging. Alternatively, cells were imaged directly in chambers with TBS.

**Western blotting.** For analysis of protein expression levels, quantitative western blots were performed using infrared fluorescence imaging on an Odyssey Imager (LI-COR). Cell homogenates were prepared in ice-cold 1X RIPA lysis buffer (Cayman Chemical) + 1X Protease/Phosphatase Inhibitor Cocktail (Cell Signaling). After one freezing cycle, lysates were spun at $18,000 \times g$ for 5 min. Protein concentration was determined by Bradford protein assay dye reagent (Bio-Rad). Aliquots of supernatants were mixed with 4× sample buffer (LI-COR) containing 10% BME, boiled for 10 min, and resolved on hand-poured sodium dodecyl sulfate (SDS)-polyacrylamide gel electrophoresis Tris-glycine gels. Proteins were transferred to a membrane in Mini Trans-Blot Cell (Bio-Rad) or Trans-Blot Turbo Transfer System (Bio-Rad), blocked 1 h in Odyssey Blocking Buffer (TBS) (LI-COR), and probed with the corresponding primary antibody (see below) overnight at 4 ˚C. Membranes were stained and imaged for total protein using Revert 700 Total Protein Stain (LI-COR) before blocking where applicable. Membranes were then rinsed with TBST 3 times for 5 min, and incubated with secondary antibodies (see below) for 1–3 h at room temperature. Membranes were rinsed again with TBST and then imaged on Odyssey Imager. Image analysis was performed using Image Studio Lite software (LI-COR). Fluorescent band intensity was normalized to GAPDH loading control or histone H3, depending on the experiment. For puromycinylated or HPG-labeled protein blots, the lane was normalized to the Revert total protein stain (LI-COR) of the corresponding region. The puromycinylated protein/total protein is normalized to the control values for individual blots. In Fig. 1e, two PBS control samples were blotted twice with separate cohorts of mice and are included without consolidation to appropriately display the variability within the PBS control samples.

Full annotated western blots are included in Supplementary Information.

**Antibodies and labels.** DI-8-ANEPPS (Thermo Fisher, D3167; 1:1 ratio to 20% Pluronic F-127, solution added 1:500)

Hoechst 33342 trihydrochloride trihydrate (Invitrogen, H3570); all experiments 1:1000

Alpha-tubulin rabbit polyclonal (Millipore Sigma, SAB3501072-100UG); IF 1:250

Tubulin (DM1A) mouse monoclonal (Cell Signaling Technology, 3873S); IF 1:1000

Desmin rabbit polyclonal (Thermo Fisher, PA5-16705); IF in NRVMs 1:500, IF in tissue sections 1:1000

Kif5b (Clone EPR10276(B)) rabbit monoclonal (Abcam, ab167429); WB 1:1500

Rps6 (clone 54D2) mouse monoclonal (Cell Signaling Technology, 2317S); IF 1:250

Puromycin (clone 12D10) mouse monoclonal (Millipore Sigma, MABE343); WB 1:2000

GAPDH rabbit polyclonal (Biolegend, Poly 6314); WB 1:2500

Histone H3 rabbit polyclonal (Abcam, ab1791); WB 1:2000

GAPDH mouse monoclonal (VWR, A01622-40); WB 1:2500

Histone H3 mouse monoclonal (Abcam, ab24834); WB 1:1000

Goat anti-rabbit IgG AF 647 (Life Technologies, A27040); IF 1:500

Goat anti-mouse IgG AF488 (Life Technologies, A11001); IF 1:1000

**Measurement of translational activity in NRVMs.** In western blotting experiments to label nascent peptides, NRVMs were labeled with puromycin (10 μg/mL, ~21 μM) for 30 min 24 h after PE/Colchicine treatment. Control cells were pretreated with cycloheximide (20 μg/mL; ~70 μM) for 30 min prior to addition of puromycin. An additional control sample that was not treated with puromycin was included. The cells were washed twice with cold PBS and lysates were collected in 1X RIPA lysis buffer (Cayman Chemical) + 1X Protease/Phosphatase Inhibitor Cocktail (Cell Signaling). Quantification of the level of puromycinylation was performed by first defining similar ROIs encompassing the sample lane between ~15 and 260 kDa in both the REVERT total protein and anti-puromycin signals. The ratio of the integrated intensity measurements of the anti-puromycin signal and the total protein was calculated. These ratios were background corrected by subtracting ratio of the No-puromycin control lane and the values were expressed relative to the DMSO control for each western blot.

In experiments using the alkynated methionine analog, L-Homopropargylglycine (HPG, Thermo Fisher) to label nascent proteins, NRVMs treated with PE/Colchicine for 24 h were washed in methionine-free RPMI based media (Thermo Fisher) containing the treatment drugs for 4 h. After methionine depletion, 100 μM of HPG was added to each dish for 30 min before the cells were washed with PBS, fixed, and permeabilized. The ClickiT plus OPP AF488 kit (Thermo Fisher) was used to fluorescently label the HPG.

**Measurement of protein degradation rate in NRVMs.** Protein degradation protocol was adapted from ref. [38] for use with HPG. Twenty-four hours after treatment with PE/Colchicine, NRVMs were methionine depleted and labeled with HPG for 4 h. After labeling, the NRVMs were washed in serum-free culture media containing excess methionine (3 mM, Sigma-Aldrich) three times at 30-min intervals. Cells were collected in 1X RIPA lysis buffer (Cayman Chemical) + 1X Protease/Phosphatase Inhibitor Cocktail (Cell Signaling) at 0, 1, 2, and 3 days post-labeling. Colchicine or PE treatment was maintained throughout the pulse/chase steps. Protein concentration was determined by Bradford protein assay dye reagent (Bio-Rad) and 25 μg of total lysate per sample were labeled with 4 nmol of IRDye 800CW Azide Infrared dye (LI-COR Biosciences) using the ClickiT Protein Reaction Buffer kit (Thermo Fisher Scientific).

**HPG labeling in ARVMs.** After treatment with DMSO, colchicine, or actinomysin inhibitors for 24 h, ARVMs were methionine depleted for 2 h and treated with HPG (100 μM) for 1 h. Afterward, the cells were washed twice with PBS, fixed in 4% paraformaldehyde (Electron Microscopy Sciences), and permeabilized with 0.1% TritonX-100. The ClickiT plus OPP AF488 kit (Thermo Fisher Scientific) was used according to manufacturer protocol to fluorescently label the HPG.

**qPCR.** Total RNA was isolated from cardiomyocytes using RNAzol RT (Molecular Research Center) following the manufacturer's instructions. Briefly, cardiomyocytes were lysed in RNAzol RT reagent. One milliliter of the lysate was combined with 0.4 ml of water and shaken vigorously for 15 s and stored for 15 min at room temperature. Samples were then centrifuged at $12,000 \times g$ for 15 min. The supernatant was removed to a new tube and mixed with one volume of isopropanol, stored at room temperature for 10 min, then centrifuged at $12,000 \times g$ for 10 min. The RNA pellet was then washed with 75% ethanol two times and solubilized in RNase-free water. RNA concentration was determined using a Nanodrop (Thermo Fisher) and 2 μg RNA was reverse transcribed using cDNA synthesis kit (TaKaRa #6110A or SuperScript IV Thermo Fisher #18091150) following manufacturer's instructions. Twenty nanograms of cDNA template was then used to conduct RT-qPCR in three technical replicates. Primer sequences provided in Supplementary Table 1.

Similarly, total RNA was isolated from flash-frozen mouse heart samples by rapid homogenization in ice-cold RNAzol RT (20 μL/mg tissue) following the manufacturer's instructions. Reverse transcription and RT-qPCR were performed as described above. QuantStudio 3 was used to collect all qPCR amplification data.

**smFISH.** Specific smFISH probe sequences are listed below. For ARVMs and NRVMs, the Stellaris RNA FISH Protocol for Adherent Cell Types (Biosearch Technologies) was followed. Stellaris FISH buffers (Biosearch Technologies) were used for all experiments. Wash Buffer A and Hybridization Buffer were made fresh

for each experiment with a new aliquot of deionized formamide. RNAse-free solutions were used whenever possible. Briefly, ARVMs were adhered to glass coverslips using MyoTak (IonOptix) and then fixed using 4% paraformaldehyde (Electron Microscopy Sciences) in RNAse-free PBS; NRVMs were cultured directly on nano-patterned coverslips (Nanosurface Biomedical) and then treated identically to ARVMs. After at least 3 h permeabilization in 70% RNAse-free ethanol (usually overnight), coverslips were transferred to Wash Buffer A for 5 min. Humidity chambers (damp paper towels covered with parafilm) were assembled in 6-well culture plates and coverslips were placed cell-side down on top of 100 mL Hybridization buffer containing the probe set(s) of interest. Coverslips were incubated in the dark at 37 °C for 16 h. After incubation, coverslips were transferred to freshly-made Wash Buffer A and incubated in the dark at 37 °C for 30 min. New Wash Buffer A was added again for 30 min, with 1:1000 Hoechst stain added for the last 10 min. Coverslips underwent a final incubation in Wash Buffer B for 5 min at room temperature before mounting on glass slides using Prolong Diamond Antifade Mountant (Thermo Fisher) and sealed with nail polish. For tissue sections, the Stellaris Protocol for Simultaneous IF + FISH in Adherent Cells was adapted (Biosearch Technologies). Attempts at polyA smFISH were performed in both formalin-fixed, paraffin-embedded (FFPE) heart tissue sections and cryo-sections. 18s rRNA smFISH was also performed in both FFPE sections and cryo-sections. However, tissue quality and overall staining quality was better and more consistent in cryo-sections, so all smFISH 18s rRNA experiments were performed in cryo-sections. Briefly, cryo-sections were removed from storage at −80 °C and immediately washed in RNase-free PBS in Coplin jar, twice for 5 min. Slides were then incubated in 70% ethanol for at least 16 h at 4 °C. Slides were carefully dried and hybridization chambers (Grace Bio-Labs) were placed around tissue sections and sealed with vacuum grease. Chambers were immediately filled with RNase-free PBS for 5 min, then pre-warmed 10 µg/ml proteinase K (Invitrogen) in RNase-free PBS for 30 min at 37 °C. Afterward, chambers were washed with RNase-free PBS twice for 5 min at room temperature prior to a 5 min incubation with Wash Buffer A. Wash Buffer A was aspirated and replaced with Hybridization Buffer containing 18s rRNA probe set and anti-desmin antibody. Slides were placed on a tray with damp paper towels and incubated at 37 °C overnight in the dark. Hybridization Buffer was then removed and the tissue was incubated with fresh Wash Buffer A twice for 30 min at 37 °C in the dark. Next, Wash Buffer B containing secondary antibody was added to the tissue for 2 h at room temperature in the dark. Hoechst stain was added for the last 10 min. Fresh Wash Buffer B was added for 5 min prior to mounting with Prolong Diamond Antifade Mountant (Thermo Fisher) and sealing with nail polish. Slides were left to cure in the dark for at least 24 h prior to imaging.

**Custom smFISH probe set sequences**. All probes were reconstituted in RNAse-free TE buffer (10 mM Tris-HCl, 1 mM EDTA, pH 8.0) and stored at a final concentration of 12.5 µM

PolyA mRNA: dT 30-mer 3′ AF 647 (IDT); smFISH 500 nM (isolated cell and tissue section) final concentration

Rat Actc1 Quasar 670: Described in Materials and methods, ref. [4], smFISH 500 nM final concentration

Rat Dsp TAMRA: Described in Materials and methods, ref. [4], smFISH 500 nM final concentration

Rat Myh6 Quasar 670: Described in Materials and methods, ref. [4], smFISH 500 nM final concentration

Rat 18s rRNA Quasar 670: Probe set sequence provided in Supplementary Table 2. smFISH (isolated cell and tissue section) 125 nM final concentration.

**Imaging and analysis**. Airyscan confocal imaging for all experiments except smFISH in NRVMs was carried out on a Zeiss 880 Airyscan confocal microscope operating on an Axiovert Z1 inverted microscope equipped with EC-Plan-Neofluar ×10 air 0.30 numerical aperature (NA), Plan-Apochromat ×20 air 0.8 NA, Plan-Apochromat ×40 oil 1.4 NA, C-Apochromat ×40 water 1.2 NA, and Plan-Apochromat ×63 oil 1.4 NA objectives. Image analysis was performed using ZEN Black software for Airyscan processing, which involves signal integration from the 32 separate sub-resolution detectors in the Airyscan detector and subsequent deconvolution of this integrated signal. For smFISH in NRVMS, widefield imaging was carried out on a Zeiss Axio Observer 7 inverted microscope using a Zeiss Axiocam 702 monochrome CMOS camera and 503 color CCD camera equipped with ×100 oil objective. Image analysis was performed using ZEN Blue software with default deconvolution (constrained, iterative). Image processing was performed using FIJI.

For cell size analysis in NRVMs (Fig. 2b, c), 4-µm z-stacks (4 slices) were acquired and cell outlines were traced using the DI-8-ANEPPS signal in the slice where each cell was in focus and closest to the substrate in order to capture the maximal area. For each condition and experiment, two 3 × 3 or 4 × 4 tiles were chosen to acquire using transmitted light in distinctly different regions of the patterned dish with a cellular confluency representative of the entire dish.

For tubulin staining in NRVMs (Fig. 2b), 2.25-µm z-stacks (10 slices) were acquired and max-intensity z-projections are shown.

For HPG staining in NRVMS (Fig. 2e), 5-µm z-stacks (5 slices) were acquired. Maximum-intensity z-projections were created and cell outlines were manually traced and quantification was averaged for each field of view. For the line scans, the Hoechst channel was used to place a 1-µm thick, 15-µm long line from the edge of the nucleus to the cell periphery.

For all smFISH staining in isolated adult cardiomyocytes (Fig. 4a, Supplementary Figs. 4d and 5a, c), 6–15-µm z-stacks (6–15 slices) were acquired, depending on thickness of the cell. Max-intensity z-projections were used to quantify mean fluorescence and are shown in the image. For N/C and ICD/C ratios, a max-intensity projection of the Hoechst channel was used to trace the nuclear outline, which was expanded by 1 µm to produce the nuclear + perinuclear (N) ROI. The Hoechst channel was also used to outline the edge of the intercalated disc and a region that extended ~3 µm into the cell was defined as the ICD. The cytosolic ROI was defined as a 5 µm-wide region directly adjacent to the ICD. For line scan analysis of Actc1 and Myh6, lines were drawn between the two nuclei, starting in the middle of each and from the middle of one nuclei to the farthest edge of the cell. Each fluorescence intensity trace was then normalized to the initial value and plotted. For each condition and experiment, cells were randomly chosen using transmitted light to image. If the coverslip was dense enough with cells, two 4 × 4 tiles were imaged, randomly chosen using transmitted light.

For Rps6 and HPG staining in isolated adult cardiomyocytes (Supplementary Fig. 4a, Fig. 4a), 10-µm z-stacks (10 slices) were imaged with the stack centered at the nuclei. A max-intensity projection of the Hoechst channel was used to trace the nuclear outline, which was expanded by 1 µm to produce the nuclear + perinuclear (N) ROI. The Hoechst channel was also used to outline the edge of the intercalated disc and a region that extended ~3 µm into the cell was defined as the ICD. The cytosolic ROI was defined as a 5-µm-wide region directly adjacent to the ICD.

For simultaneous smFISH and IF in tissue sections (Fig. 4d, e), 2-µm z-stacks (5 slices) were acquired. A max-intensity projection of the Hoechst channel was used to identify both perinuclear (N) and cytosolic (C) ROIs, and a max-intensity projection of the desmin channel was used to identity intercalated disc (ICD) ROIs. Example ROIs are shown in Supplementary Fig. 6A. Eight to nine random rectangular cytosolic ROIs were chosen in regions in the center of cells, far from nuclei, and averaged to calculate mean cytosolic 18s rRNA fluorescence for each field of view. ICD ROIs encompass a 2 µm border on each side of the desmin ICD signal. A max-intensity projection of the 18s rRNA channel was then used to measure mean fluorescence intensity using the previously identified ROIs. Imaged regions were chosen using transmitted light to identify areas of the tissue with well-striated and uniform structure.

For smFISH in NRVMs (Fig. 5a, c), 4.6-µm z-stacks (20 slices) were acquired. For each condition and experiment, two 2 × 4 or 2 × 3 tiles were chosen to acquire using transmitted light in distinctly different regions of the patterned dish with a cellular confluency representative of the entire dish. Max-intensity projections of the most in-focus 11 slices were used to measure mean fluorescence for Actc1 and number of puncta for Dsp. For line scan analysis of Actc1, lines of "30" thickness in Fiji were drawn from the middle of each mono-nucleated cell to the farthest edge of the cell, using a composite of the Hoechst and Actc1 channels. Each was normalized to the initial value. The mean values of 1 µm were binned every 5 µm and plotted, e.g., "0 µm" is the mean value from 0 to 0.99 µm, "5 µm" is the mean value from 5 to 5.99 µm, etc. For puncta analysis of Dsp, the Hoechst channel was used to outline the nuclei of each mono-nucleated cell. A composite of the Actc1 and Hoechst channels was used to outline the total area of the cell. The outline of the total cell area was "enlarged" by −2 µm to demarcate the periphery and cytosol of each cell. Then, the Dsp channel was background-subtracted using a 50-pixel rolling ball radius and thresholded. Particles were then counted using particle analysis for each cell using the previously identified nuclear, cytosolic and peripheral ROIs.

**Statistics and data handling**. Statistical analysis was performed using OriginPro (Version 9 and 2018). Statistical test and information on biological and technical replicates can be found in the figure legends. Degrees of freedom, F values, and t-values are reported in Supplementary Table 1.

For box plots, the mean line is shown, with whiskers denoting standard error (SE) from the mean. Statistical tests for each comparison are denoted in the figure legends.

**Reporting summary**. Further information on research design is available in the Nature Research Reporting Summary linked to this article.

## Data availability

Source data are provided with this paper. Additional relevant data are available at reasonable request. Source data are provided with this paper.

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

## Acknowledgements

We thank the Penn Center for Musculoskeletal Disorders Histology Core (P30-AR069619) and the Cell and Developmental Biology Microscopy Core at the UPenn Perelman School of Medicine. Funding for this work was provided by the National Institutes of Health (NIH) R01s-HL133080 and HL149891 to B.P., T32 HL007843 to K.U., T32 AR053461 to E.S., Medical Scientist Training Program T32 GM07170 to N.E., by the U.S. Israel Binational Science Foundation (BSF) Award 2019126 to B.P. and I.K., by the Fondation Leducq Research grant no. 20CVD01 to B.P. and I.K., and by the Center for Engineering MechanoBiology to B.P. and E.S. through a grant from the National Science Foundation's Science and Technology Center program: 15-48571.

## Author contributions

B.P. and I.K. conceived the project and contributed to its design. B.P., E.A.S., and K.U. designed experiments. E.A.S., K.U., M.V., M.I., S.P., and A.B. performed experiments. B.P., E.A.S., K.U., M.V., and M.I. analyzed data. B.P., E.A.S., K.U., and N.E. prepared the manuscript.

## Competing interests

The authors declare no competing interests.
