## [Peer Review File · Nature Communications]

REVIEWERS' COMMENTS

Reviewer #2 (Remarks to the Author):

This is a revised version of a manuscript that was originally submitted to Nature. The authors adequately addressed initial concerns, and the manuscript is much improved. The findings define a novel regulatory pathway that controls cardiomyocyte hypertrophy. The data are convincing and the manuscript is very well written. It is likely to be of great interest to the readership of Nature Communications.

Reviewer #3 (Remarks to the Author):

The authors have addressed my concerns with additional experimentation and where appropriate, have added thoughtful discussion. I have no further suggestions.

Response to Reviewer Comments:

Reviewer #2 (Remarks to the Author):

This is a revised version of a manuscript that was originally submitted to Nature. The authors adequately addressed initial concerns, and the manuscript is much improved. The findings define a novel regulatory pathway that controls cardiomyocyte hypertrophy. The data are convincing and the manuscript is very well written. It is likely to be of great interest to the readership of Nature Communications.

Thank you for your comments and we appreciate the time you have taken to review our manuscript.

Reviewer #3 (Remarks to the Author):

The authors have addressed my concerns with additional experimentation and where appropriate, have added thoughtful discussion. I have no further suggestions.

Thank you for your remarks and thank you for reviewing our manuscript.